

# Multifidelity Monte Carlo Estimation for Efficient Uncertainty Quantification in Climate-Related Modeling

Anthony Gruber [1], Max Gunzburger [1,2], Lili Ju [3], Rihui Lan [3], and Zhu Wang [3]

[1]Department of Scientific Computing, Florida State University, Tallahassee, FL 32306, USA.
[2]Oden Institute for Engineering and Sciences, University of Texas, Austin, TX 78712, USA.
[3]Department of Mathematics, University of South Carolina, Columbia, SC 29208, USA.

**Correspondence:** Anthony Gruber (agruber@fsu.edu)

**Abstract.** Uncertainties in an output of interest that depends on the solution of a complex system (e.g., of partial differential equations with random inputs) are often, if not nearly ubiquitously, determined in practice using Monte Carlo (MC) estimation. While simple to implement, MC estimation fails to provide reliable information about statistical quantities (such as the expected value of the output of interest) in application settings such as climate modeling for which obtaining a single realization of the output of interest is a costly endeavor. Specifically, the dilemma encountered is that many samples of the output of interest have to be collected in order to obtain an MC estimator having sufficient accuracy; so many, in fact, that the available computational budget is not large enough to effect the number of samples needed. To circumvent this dilemma, we consider using multifidelity Monte Carlo (MFMC) estimation which leverages the use of less costly and less accurate surrogate models (such as coarser grids, reduced-order models, simplified physics, interpolants, etc.) to achieve, for the same computational budget, higher accuracy compared to that obtained by an MC estimator or, looking at it another way, an MFMC estimator obtains the same accuracy as the MC estimator at lower computational cost. The key to the efficacy of MFMC estimation is the fact that most of the required computational budget is loaded onto the less costly surrogate models, so that very few samples are taken of the more expensive model of interest. We first provide a more detailed discussion about the need to consider an alternate to MC estimation for uncertainty quantification. Subsequently, we present a review, in an abstract setting, of the MFMC approach along with its application to three climate-related benchmark problems as a proof-of-concept exercise.

## 1 Introduction

In many application settings — climate modeling being a prominent one — large computational costs are incurred when solutions to a given model are approximated to within an acceptable accuracy tolerance. In fact, this cost can be prohibitively large when one has to obtain the results of multiple simulations, as is the case for, e.g., uncertainty quantification, control and optimization, to name a few. Thus, there is often a need for compromise between the accuracy of simulation algorithms and the number of simulations needed to obtain, say, in the uncertainty quantification setting, accurate statistical information.

For example, consider the following case which represents the focus of this paper. Suppose one has a complex system, say, a system of discretized partial differential equations, for which the input data depends on a vector of randomly distributed parameters $\mathbf{z}$. Letting $\mathbf{u}(\mathbf{z})$ denote the solution of the discretized partial differential equations, we define an output of interest



$F\big(\mathbf{u}(\mathbf{z})\big) = F(\mathbf{z})$ that depends on $\mathbf{u}(\mathbf{z})$ so that, of course, it also depends on the choice of $\mathbf{z}$. Next, suppose we wish to use Monte Carlo sampling to estimate the expected value $Q = \mathbb{E}[F(\mathbf{z})]$ of the output of interest $F(\mathbf{z})$, i.e., we have the Monte Carlo estimator of $Q$ given by

$$Q^{MC} = \frac{1}{M} \sum_{m=1}^{M} F(\mathbf{z}_m) = \frac{1}{M} \sum_{m=1}^{M} F\big(\mathbf{u}(\mathbf{z}_m)\big) \approx \mathbb{E}[F(\mathbf{z})] = Q, \tag{1}$$

where $\{\mathbf{z}_m\}_{m=1}^{M}$ denotes a set of $M$ independent and identically distributed samples of the random vector $\mathbf{z}$. This is a common
task in uncertainty quantification which provides useful statistical information in both predictive and inferential applications.

On the other hand, Monte Carlo estimation is not without its drawbacks. Consider the following scenario. Let $\delta < 1$ denote a measure of the spatial grid size used to discretize the PDE system in question, and suppose that $\delta$ is normalized by the diameter of the domain in which that system is posed. Then, we have that the error in the approximate solution is of $\mathcal{O}(\delta^{\nu_d})$ for some $\nu_d > 0$ and that the cost (e.g., measured in seconds, or days, or weeks) of a simulation which obtains the approximate
solution is of $\mathcal{O}(\delta^{-\nu_c})$ for some $\nu_c > 0$. Neglecting any additional costs connected with the evaluation of $F(\mathbf{z}) = F\big(\mathbf{u}(\mathbf{z})\big)$ (for any given $\mathbf{z}$) once $\mathbf{u}(\mathbf{z})$ is obtained, it is clear that the cost $C$ incurred when obtaining the Monte Carlo estimator $Q^{MC}$ is of $\mathcal{O}(M\delta^{-\nu_c})$. Of course, it is well known that the accuracy of a Monte Carlo estimator is of $\mathcal{O}(1/\sqrt{M})$ so that in order for the accuracy of the estimator to be commensurate with the discretization error, we must have that $1/\sqrt{M} \propto \mathcal{O}(\delta^{\nu_d})$. Therefore, the needed number of samples is proportional to $M \propto \delta^{-2\nu_d}$, and the *total cost $CM$ incurred when determining the Monte Carlo*
*estimator $Q^{MC}$ is of $\mathcal{O}(\delta^{-2\nu_d - \nu_c})$.*

The cost $CM$ can quickly get out of control when dealing with large-scale problems. For example, suppose that the approximate solution of the PDE system is second-order accurate, i.e., $\nu_d = 2$, and that a single simulation incurs a cost of $\mathcal{O}(\delta^{-3})$ (i.e., $\nu_c = 3$) which is the best-case scenario in three dimensions. We then have that the number of samples needed is $M \propto \delta^{-4}$ and the total cost $CM \propto \delta^{-7}$. Thus, with even a modest grid size of $\delta = 0.01$ we have that number of samples needed $M \propto 10^8$
and the total cost $CM \propto 10^{14}$, which is too high for practical use. Turning things around, suppose instead that the available computational budget allows for at most ten thousand simulations, i.e., $M = 10^4$, so that the resulting accuracy of the Monte Carlo simulator is of $\mathcal{O}(10^{-2})$. Here, the estimator error and approximation error are already commensurate when $\delta = 0.1$, making the choice of a smaller delta redundant if not infeasible. In fact, in this case there is no way to achieve the four digits of accuracy sought by choosing $\delta = 0.01$, since the best we can do with the available budget is on the order of $\mathcal{O}(10^{-2})$. Note
that the situation becomes even worse when the PDE system in question is time dependent, as a single simulation incurs an even larger cost when one accounts for the number of time steps used in the simulation.

Given that the cost of MC estimation is at times prohibitively expensive, it comes as no surprise that many alternatives or run-arounds to such estimation have been proposed. One approach in this direction has led to the development of many different random parameter sampling schemes, e.g., quasi-Monte Carlo sampling, sparse-grid sampling, importance sampling, Latin
hypercube sampling, lattice sampling, compressed sensing, to name just a few, for which the estimation error is guaranteed to be smaller than its Monte Carlo equivalent; see, e.g., Addcock et. al. (2022); Evans and Swartz (2000); Gunzburger et. al. (2014); Nierderreiter (1992); Sloan (1994); Smith (2013). On the other hand, some of these alternate approaches require





smoothness of solutions to achieve better accuracy. Furthermore, most, if not all, of these methods are superior to Monte Carlo sampling only for moderate dimension of the parameter vector $\mathbf{z}$; see the references just cited.

A second approach towards reducing the cost of Monte Carlo (and for that matter for any type of) uncertainty estimation is to use approximate solutions of the PDE system that are less costly to obtain compared to the cost of obtaining the approximation of actual interest. For example, using simulations obtained using coarser grids or using reduced-order models such as reduced-basis or proper orthogonal decomposition methods are less costly as are interpolation and support vector machine approximations; see, e.g., Cristianini and Shawe-Taylor (2000); Fritzen and Ryckelynck (2019); Keiper et. al. (2018); Quarteroni

and Rozza (2014); Quarteroni et. al. (2016); Steinwart and Christmann (2008). However, such approaches, by definition, result in less accurate approximations compared to the accuracy that one wants to achieve.

    In this paper we do not consider any of the possible alternate sampling schemes nor do we exclusively consider using less costly and less accurate approximate solutions of the PDE system. Instead, because of the near ubiquity of its use in practice, *our goal is to outperform traditional Monte Carlo estimation by using a nontraditional Monte Carlo sampling strategy* and

in so doing *refrain from incurring any loss of accuracy*. To meet this goal, we invoke *multifidelity Monte Carlo estimation* which, *in addition to the expensive and accurate PDE system approximation of interest* (hereafter referred to as the "truth" approximation) *also uses cheaper to obtain and less accurate approximations* (which are referred to as the *"surrogates"*). The bottom line is that multifidelity Monte Carlo estimation meets our goal by leveraging increased sampling of the less accurate/less costly approximations alongside low sampling of the more expensive/more accurate truth approximation. The

multifidelity Monte Carlo algorithm systematically determines the number of samples taken from each surrogate (i.e., there is no guess work involved) and systematically (i.e., again no guess work involved) combines the samples of the surrogates to obtain the desired estimator. We note that multifidelity Monte Carlo estimation has already been shown to outperform Monte Carlo estimation in a variety of application settings; see, e.g., Clare et. al. (2022); Dimarco et. al. (2022); Konrad (2019); Law et. al. (2022); Modderman (2021); Quick et. al. (2019); Rezaeiravesh et. al. (2020); Romer et. al. (2020); Valero et. al. (2022)

for some examples.

    In Section 2, we review how the Monte Carlo and multifidelity Monte Carlo estimators are constructed in an abstract setting; here we follow the expositions in Peherstorfer et. al. (2016) and also in Gruber et. al. (2022). Then, in Sections 3 and 4 we demonstrate the effectiveness of multifidelity Monte Carlo estimation using three well-known climate-related benchmarks. Sections 3.1 and 3.2 are respectively devoted to single layer shallow-water equations model for the SOMA Test Case of

Wolfram et. al. (2015) and Test Case 5 of Williamson et. al. (1992). Section 4 is devoted to a benchmark case for the first-order model for ice sheets; see Blatter (1995); Pattyn (2003); Perego et. al. (2012); Tezaur et. al. (2015). We close by providing some concluding remarks in Section 5.

## 2   Monte Carlo and multifidelity Monte Carlo estimators

An abstraction of the specific settings considered in Sections 3 and 4 involves first





– having in hand a (*discretized*) *partial differential equation* (PDE) system for which the solution $\mathbf{u}(\mathbf{z})$ depends on a random vector of parameters $\mathbf{z} \in \Gamma$, where $\Gamma$ denotes a parameter domain.

Note that the input data to this PDE system, e.g., forcing terms, initial conditions, coefficients, etc., could depend on one or more of the components of the random vector $\mathbf{z}$. Moreover,

     – we are interested in situations such that, *for any $\mathbf{z} \in \Gamma$, obtaining $\mathbf{u}(\mathbf{z})$ is a costly endeavor.*

In addition,

     – we define a scalar *output of interest* (OoI) $F^1(\mathbf{z}) = F^1\big(\mathbf{u}(\mathbf{z})\big)$ that depends on the solution $\mathbf{u}(\mathbf{z})$ of the (discretized) PDE system; of course, if obtaining $\mathbf{u}(\mathbf{z})$ is costly, then so is obtaining $F^1(\mathbf{z})$.

While they are not considered in this work, vector-valued outputs of interest can also be treated with multifidelity Monte Carlo techniques. OoIs could be, e.g., averages or extremal values of the energy, etc., associated with the solution $\mathbf{u}(\mathbf{z})$.

Having defined an OoI $F^1(\mathbf{z})$,

     – we let $Q_1 = \mathbb{E}\big[F^1\big]$ denote the *quantity of interest* (QoI) corresponding to the $F^1(\mathbf{z})$,

where $\mathbb{E}[\cdot]$ denotes the expected value with respect to $\mathbf{z}$. Because the estimation of $Q_1 = \mathbb{E}\big[F^1\big]$ is the central goal of this paper,

     – we refer to $F^1(\mathbf{z})$ as the *"truth" output of interest.*

Commonly, even ubiquitously,

     – a Monte Carlo (MC) sampling method is used to (approximately) quantify the uncertainty in the chosen OoI $F^1(\mathbf{z})$.

Specifically,

     – MC sampling is used to estimate the quantity of interest $Q_1 = \mathbb{E}\big[F^1\big]$ corresponding to $F^1(\mathbf{z})$,

i.e., we have that

– the MC estimator of the QoI $Q_1 = \mathbb{E}\big[F^1\big]$ is given by

$$Q_1^{MC} = \frac{1}{M_1} \sum_{m=1}^{M_1} F^1(\mathbf{z}_m) \approx Q_1 = \mathbb{E}\big[F^1\big], \tag{2}$$

where $\{\mathbf{z}_m\}_{m=1}^{M_1}$ denotes a set of $M_1$ randomly selected points in the parameter domain $\Gamma$.

We are then faced with the following dilemma: on the one hand,

     – obtaining an acceptably accurate MC estimator $Q_1^{MC}$ of the QoI $Q_1$ requires obtaining the solution $\mathbf{u}(\mathbf{z})$ of the discretized

115          PDE system at many randomly selected points $\mathbf{z} \in \Gamma$,

and, on the other hand,

     – each of those approximate solutions $\mathbf{u}(\mathbf{z})$ are computationally costly to obtain.

Quantifying uncertainties in climate system settings are victimized by this two-headed dilemma to the extent that, e.g., accurate long-time integrations can often not be realized in practice.

Due to the issue of prohibitive computational cost, we turn to *multifidelity Monte Carlo* (MFMC) methods for uncertainty quantification. MFMC methods leverage the availability of surrogate outputs of interest $F^k(\mathbf{z})$, $k = 2, \ldots, K$, which have smaller computational complexity compared to that of the truth OoI $F^1(\mathbf{z})$. As mentioned in Section 1, there are many types





of surrogates that can be used for this purpose, e.g., discretized PDEs with coarser spatial grids and larger time steps, reduced-basis and proper orthogonal decomposition-based reduced-order models, and interpolants of $F^1(\mathbf{z})$, to name just a few.

For the truth and the $K-1$ surrogate OoIs, Monte Carlo (MC) sampling yields the $K$ unbiased estimators

$$Q_k^{MC} = \frac{1}{M_k} \sum_{m=1}^{M_k} F^k(\mathbf{z}_m) \approx \mathbb{E}\left[F^k\right] = Q_k \quad \text{for } k = 1, \dots, K, \tag{3}$$

where $\mathbf{z}_1, \dots, \mathbf{z}_{M_k}$ denote $M_k$ i.i.d. samples of $\mathbf{z} \in \Gamma$. The goal of an MFMC method is using an appropriate linear combination of the $K$ MC estimators $Q_k^{MC}$ in (3) to estimate the QoI $Q_1 = \mathbb{E}\left[F^1\right]$. Specifically, we define the MFMC estimator as

$$Q^{MFMC} = Q_1^{MC} + \sum_{k=2}^{K} \alpha_k \left(Q_k^{MC} - Q_{k-1}^{MC}\right) \approx Q_1 = \mathbb{E}\left[F^1\right], \tag{4}$$

where $\{\alpha_k\}_{k=2}^{K}$ denotes a sequence of scalar weights and $\{M_k\}_{k=1}^{K}$ denotes a nondecreasing sequence of integers defining the numbers of samples.

Letting $C_k$ denote the cost of evaluating the $k^{th}$ output of interest $F^k$, the costs of computing the respective MC and MFMC estimators are given as

$$C_k^{MC} = C_k M_k \quad \text{for } k = 1, \dots, K \qquad \text{and} \qquad C^{MFMC} = \sum_{k=1}^{K} C_k^{MC} = \mathbf{C} \cdot \mathbf{M},$$

where $\mathbf{C} = \{C_k\}_{k=1}^{K}$ and $\mathbf{M} = \{M_k\}_{k=1}^{K}$ denote $K$-vectors formed from the previously introduced sequences.

The variance $\sigma_k^2$ of the $k^{th}$ output of interest $F^k$ and the correlation $\zeta_{k,k'}$ between the outputs of interest $F^k$ and $F^{k'}$ are given by, for $k', k = 1, \dots, K$,

$$\sigma_k^2 = \text{Var}\left[F^k(\mathbf{z})\right] \qquad \text{and} \qquad \zeta_{k,k'} = \frac{\text{Cov}\left[F^k(\mathbf{z}), F^{k'}(\mathbf{z})\right]}{\sigma_k \sigma_{k'}},$$

respectively, where $\text{Cov}\left[\cdot, \cdot\right]$ denotes the statistical covariance. We then have that the *mean-squared error* (MSE) incurred by

the MC estimator $Q_k^{MC}$ of the QoI $Q_k$ is given by

$$\begin{aligned} e(Q_k^{MC}) &= \mathbb{E}\left[\left(Q_k - Q_k^{MC}\right)^2\right] = \mathbb{E}\left[\left(\mathbb{E}\left[F^k(\mathbf{z})\right] - Q_k^{MC}\right)^2\right] \\ &= \mathbb{E}\left[F^k(\mathbf{z})\right]^2 - \mathbb{E}\left[F^k(\mathbf{z})^2\right] = \frac{\sigma_k^2}{M_k} \qquad \text{for } k = 1, \dots, K \end{aligned} \tag{5}$$

whereas the MSE incurred by the $Q^{MFMC}$ estimator of the QoI $Q_1$ is given by (see Peherstorfer et. al. (2016))

$$e(Q^{MFMC}) = \frac{\sigma_1^2}{M_1} + \sum_{k=2}^{K} \left(\frac{1}{M_{k-1}} - \frac{1}{M_k}\right) \left(\alpha_k^2 \sigma_k^2 - 2\alpha_k \sigma_k \sigma_1 \zeta_{1,k}\right). \tag{6}$$

Note that it can be shown (see e.g. Peherstorfer et. al. (2016)) that the MSE for $Q^{MFMC}$ is lower than that for $Q_1^{MC}$ if and

only if

$$\sqrt{\frac{e(Q^{MFMC})}{e(Q_1^{MC})}} = \sum_{k=1}^{K-1} \sqrt{\frac{C_k}{C_1}\left(\zeta_{1,k}^2 - \zeta_{1,k+1}^2\right)} < 1. \tag{7}$$





Given a fixed computational budget $B$, MFMC aims to construct an optimal sampling strategy $\mathbf{M} = \{M_k\}_{k=1}^K$ along with an optimal set of weights $\boldsymbol{\alpha} = \{\alpha_k\}_{k=2}^K$ so that the MSE (6) of the multifidelity estimator $Q^{MFMC}$ is lower than the MSE (5) of the Monte Carlo estimator $Q_1^{MC}$. Viewed differently, this means that an appropriate estimator $Q^{MFMC}$ can achieve a fixed

MSE $\varepsilon > 0$ at a smaller computational cost compared to that incurred for the Monte Carlo estimator $Q_1^{MC}$ achieving MSE $\varepsilon$.

In Peherstorfer et. al. (2016, 2018), unique optimal values of $\mathbf{M} = \{M_k\}_{k=1}^K$ and $\boldsymbol{\alpha} = \{\alpha_k\}_{k=2}^K$ are analytically obtained by minimizing the MSE (6) of the $Q^{MFMC}$ estimator (4) *over the real numbers*. However, the values $\mathbf{M}$ must certainly be integers for practical use, and the heuristic of Peherstorfer et. al. (2016) which determines the optimal values can result in either a biased estimator of the expectation or (with naïve modification) a violation of the given budget $B$. For "small" computational

budgets, the consequences of this can be quite severe as illustrated in Gruber et. al. (2022).

Because "small" computational budgets are of high interest for climate modeling, here, instead of using the MFMC method of Peherstorfer et. al. (2016, 2018), we use the modified MFMC method of Gruber et. al. (2022) which guarantees that the optimal sampling numbers are integers and that the computational budget is not exceeded. In that method, instead of simply minimizing the MSE $e(Q^{MFMC})$ defined in (6), the modified MFMC estimator is determined through the use of at least some

of the sequential minimization problems given by

for $k = 1, 2, \ldots, K$ and, if $k > 1$, for given $M_{1,1}, \ldots, M_{k-1,k-1}$,
minimize the functional

$$\mathcal{L}^k(\mathbf{M}_k, \boldsymbol{\alpha}_k; \lambda_k, \boldsymbol{\mu}_k, \xi_k) = \tag{8}$$

$$e(Q^{MFMC}) + \lambda_k \left( \sum_{k'=k}^K C_{k'} M_{k',k} - \left( B - \sum_{k'=1}^{k-1} C_{k'} \right) \right) + \sum_{k'=k+1}^K \mu_{k',k}\big(M_{k',k} - M_{k'-1,k}\big) - \xi_k M_{k,k}$$

where $\mu_{k',k}$ for $k' = k+1, \ldots, K$, $\lambda_k$, and $\xi_k > 0$ are Lagrange multipliers.

In (8), the first term added to the MSE $e(Q^{MFMC})$ enforces the budget constraint, or, more precisely, for each $k$, enforces that the remaining available budget suffices to not exceed the given budget $B$ after that budget has been depleted by the

sampling already effected for the OoIs $F^{k'}$, $k' = 1, \ldots, k-1$, where, of course, for $k = 1$ the whole budget $B$ is available. The second term added in (8) enforces the monotone non-decrease in the numbers of samples $M_{k',k}$ for $k' > k$. The third term added to the MSE enforces the positivity of $M_{k,k}$.

Repeating the arguments made in Peherstorfer et. al. (2016) concerning optimal choices for the sample numbers and weights, the results given in the box **A** below are proved in Gruber et. al. (2022).

**Remark 2.1.** It is clear from (11) that the weights $\alpha_{k',k}^*$ depend only on input data, specifically on the variances and correlations of the OoIs $\{F^{k'}\}_{k'=1}^K$. As such the value of $\alpha_{k',k}^*$ is independent of $k$ so that

$$\alpha_{k',k}^* = \frac{\zeta_{1,k'}\sigma_1}{\sigma_{k'}} = \alpha_{k',1}^* \qquad \text{for all } k = 1, \ldots, K \text{ and all } k' = k+1, \ldots, K.$$

Thus, the set of weights $\{\alpha_{k',1}^*\}_{k'=2}^K$ determined from the minimization of $\mathcal{L}^1$ suffices to determine $\alpha_{k',k}^*$ for all such $k'$ and $k$, i.e., all $\alpha_{k',k}^*$ for all $k'$ and $k$ are determined once and for all by the minimization of $\mathcal{L}^1$. □





---

**A. Optimal non-integer sampling numbers for the modified MFMC method;** see Gruber et. al. (2022)

- Let $\{F^k\}_{k=1}^K$ denote the set of computational outputs of interest with correlation coefficients $\{\zeta_{1,k}\}_{k=2}^K$ and computational costs $\{C_k\}_{k=1}^K$ that respectively satisfy

$$|\zeta_{1,k-1}| > |\zeta_{1,k}| \quad \text{and} \quad \frac{C_{k-1}}{C_k} > \frac{\zeta_{1,k-1}^2 - \zeta_{1,k}^2}{\zeta_{1,k}^2 - \zeta_{1,k+1}^2}, \qquad k = 2, \ldots, K. \tag{9}$$

The first requirement is easily satisfied by a reordering of the of the surrogate outputs of interest $F^2, \ldots, F^K$. If after that reordering, the second requirement is not satisfied for some $k \in \{2, \ldots, K\}$, then that $F^k$ is removed from the list of surrogates.

- Let $M_{1,1}, \ldots, M_{k-1,k-1}$ be given which are all positive by construction.

- Also let $\mathbf{M}_k^* = \{M_{k',k}^*\}_{k'=k}^K$ and $\boldsymbol{\alpha}_k^* = \{\alpha_{k',k}^*\}_{k'=k+1}^K$ and let

$$r_{k',k}^* = \frac{M_{k',k}^*}{M_{k,k}^*} = \sqrt{\frac{C_k}{C_{k'}} \left( \frac{\zeta_{1,k'}^2 - \zeta_{1,k'+1}^2}{\zeta_{1,k}^2 - \zeta_{1,k+1}^2} \right)}, \qquad k' = k, \ldots, K.$$

- Then, for each $k = 1, \ldots, K$, the *unique global minimizer* $(\mathbf{M}_k^*, \boldsymbol{\alpha}_k^*)$ of the functional $\mathcal{L}^k$ is given by

$$M_{k,k}^* = \frac{B - \sum_{k'=1}^{k-1} C_{k'}}{\sum_{k'=k}^K C_{k'} r_{k',k}}, \tag{10}$$

and

$$M_{k',k}^* = M_{k,k}^* r_{k',k}^* \quad \text{and} \quad \alpha_{k',k}^* = \frac{\zeta_{1,k'} \sigma_1}{\sigma_{k'}}, \qquad k' = k+1, \ldots, K. \tag{11}$$

---

Unfortunately, as was the case in Peherstorfer et. al. (2016, 2018), the results given in box **A** do not immediately lead to a practical MFMC method because the sample numbers $M_{k',k}^*$ given in (10) and (11) are not, in general, integers. Moreover, as was also the case in Peherstorfer et. al. (2016, 2018), the first and perhaps even the first few sampling numbers at stage $k' = 1$ may be $< 1$ in addition to being noninteger. In those papers, a rounding procedure is implemented, i.e., the sample numbers $M_{k',1}$ are replaced by integers, though this is unsuitable for scenarios where $M_{1,1} < 1$ as the choice $M_{1,1} = 0$ leads to a biased

estimator while the choice $M_{1,1} = 1$ exceeds the computational budget. On the other hand, the modified MFMC method in box **B** below is constructed to avoid these issues while preserving the optimality of the original solution (up to some amount of rounding). Note that in that box and elsewhere, $\lfloor \cdot \rfloor$ denotes rounding downwards to the nearest integer.





---

**B. Practical near-optimal integer sampling numbers for the modified MFMC method;** see Gruber et. al. (2022)

---

For $k = 1, 2, \ldots,$

    **Ia**. if the minimization of the Lagrangian functional $\mathcal{L}^k$ constrained by $M_{1,1} = M_{2,2} = \cdots = M_{k-1,k-1} = 1$ results in all $K - k + 1$ members of the set $\{M_{k',k}\}_{k'=k}^{K}$ satisfying $M_{k',k} \geq 1$, $k' = k, \ldots, K$.

Then,

    **Ib**. simply rounding downwards to the nearest integer produces integer sample numbers $M_{k',k}^* \to \lfloor M_{k',k}^* \rfloor$, $k' = k, \ldots, K$, so that the final set $\{M_{k',k'} = 1\}_{k'=1}^{k-1} \cup \{\lfloor M_{k',k}^* \rfloor\}_{k'=k}^{K}$, of sampling number optimal under the constraint $M_{1,1} = \cdots = M_{k',k'} = 1$ and *preserves the computational budget*. We then set $\widehat{K} = k - 1$ and exit to step **IV**.

Otherwise,

    **IIa**. if $M_{k,k}^* < 1$, then it is rounded upwards $M_{k,k}^* \to 1$ and the function $\mathcal{L}^{k+1}$ is minimized to obtain the remaining $K - k$ components of $M_{k',k}^*$, $k' = k+1, \ldots, K$.

This yields

    **IIb**. a new set $\{M_{1,1}^* = 1, M_{2,2}^* = 1, \ldots, M_{k,k}^* = 1\} \cup \{M_{k',k}^*\}_{k'=k+1}^{K}$ which may or may not contain entries less than 1. If it does not, the model evaluation vector becomes $\mathbf{M}^* = \{M_{k',k'} = 1\}_{k'=1}^{k} \cup \{\lfloor M_{k',k}^* \rfloor\}_{k'=k+1}^{K}$. We then set $\hat{K} = k$ and exit to step **IV**.

Otherwise,

    **III**. we increment $k \to k + 1$ and return to step **Ia**.

    **IV**. The process terminates and the final set of sample numbers is given by $\{M_{1,1}^* = 1, M_{2,2}^* = 1, \ldots, M_{\widehat{K},\widehat{K}}^* = 1\} \cup \{M_{k',\widehat{K}+1}^*\}_{k'=\widehat{K}+1}^{K}$.

---

## 3 Tests for the single-layer rotating shallow-water equations

Consider the single-layer rotating shallow-water equations (RSWEs) posed on the domain $\Gamma \times [0, T]$ and given by (see Wallis (2012))

$$\frac{\partial h}{\partial t} + \nabla \cdot (h\mathbf{u}) = 0,$$

$$\frac{\partial \mathbf{u}}{\partial t} + (\mathbf{k} \cdot \nabla \times \mathbf{u} + f)(\mathbf{k} \times \mathbf{u}) + \nabla\Big(\frac{|\mathbf{u}|^2}{2} + \rho + h_b\Big) = G(h, \mathbf{u}),$$

(12)

where

    $\Gamma$         denotes the surface of a sphere or a subset of that surface,

    $[0, T]$    denotes a time interval,

    $\mathbf{k}$         denotes a unit vector perpendicular to the surface of the sphere,

    $h(\mathbf{x}, t)$  denotes the fluid thickness,

    $\mathbf{u}(\mathbf{x}, t)$  denotes the vector velocity field tangential to the surface of the sphere,

    $G(h, \mathbf{u})$ denotes the a forcing term that depends on the specific setting,

    $f$         denotes the Coriolis parameter,





$\rho$          denotes the density,

    $h_b(\mathbf{x},t)$ denotes the bottom topography,

    $g$          denotes the (constant) acceleration due to gravity,

    $\nabla$          denotes the tangential gradient, i.e., $\nabla f = Df - (Df \cdot \mathbf{k})\mathbf{k}$ where $D$ is the derivative operator of $\mathbb{R}^3$.

Note that the expression $(\mathbf{k} \cdot \nabla \times \mathbf{u} + f)/h$ appearing in the velocity equation is known as the potential vorticity; see Wallis

(2012). Supplementing (12) are the initial conditions $h = h_0$ and $\mathbf{u} = \mathbf{u}_0$ at $t = 0$ and, if $\Gamma$ is a strict subset of the surface of the sphere, the boundary condition $\mathbf{u} \cdot \mathbf{n} = 0$ on the boundary $\partial\Gamma$ of $\Gamma$, where $\mathbf{n}$ denotes the unit vector tangent to the surface of the sphere and also (outward) perpendicular to $\partial\Omega$.

The RSWEs represent a useful simplification of the primitive equations (Wallis (2012)) which are commonly used in oceanic and atmospheric modeling and which are obtained by assuming a small ratio between the vertical and horizontal length scales.

In this way, the single-layer RSWEs describe the motion of a thin layer of fluid which lies on a rigid surface, yielding conditions used extensively in the modeling of oceanic and atmospheric flows.

Spatial discretization of the system (12) is effected using the TRiSK scheme (Ringler et. al. (2010); Thuburn et. al. (2009)) which is a staggered C-grid mimetic finite difference/finite volume scheme that preserves desirable physical properties including conservation laws for mass, energy, and potential vorticity. TRiSK discretization involves the approximation $h_\ell$ of

the height $h$ at the center of a grid cell $P_\ell$, the approximation $u_e$ of the normal to the edge component $\mathbf{u} \cdot \mathbf{n}$ of velocity at the centers of the edges of $P_\ell$, and the approximation of the potential vorticity $(\mathbf{k} \cdot \nabla \times \mathbf{u} + f)/h$ at the vertices of $P_\ell$. The necessary meshing is done using spherical centroidal Voronoi tessellation (SCVT) grids (Jacobsen et. al. (2013); Ringler et. al. (2008); Yang et. al. (2018)); examples of such grids are given in Sections 3.1 and 3.2 for the specific settings of those sections.

Temporal discretization of the system (12) is effected using an explicit $4^{th}$-order Runge-Kutta method, although many

alternative time-stepping schemes have also been proposed for this purpose; see, e.g., Leng et. al. (2019); Meng et. al. (2020); Trahan and Dawson (2012). We denote by $\{t_n\}_{n=0}^{N_t}$ with $t_0 = 0$ and $t_{N_t} = T$ the time instants used for temporal discretization.

## 3.1 A SOMA test case for the RSWE system

The specific setting considered here is the benchmark test case referred to as "simulating ocean mesoscale activity" (SOMA) (Wolfram et. al. (2015)) which involves a geodesic basin with radius 1250km centered at latitude/longitude $(35°, 0°)$ on the

surface of the Earth. Inside the basin, the depth of fluid varies from 2500m at the center to 100m on the coastal shelf, creating a realistic topography which yields interesting dynamical behaviors which are useful for studying the propagation of fronts and eddies.

In (12),

$$G(h, \mathbf{u}) = d_{\text{bottom}}(h, \mathbf{u}) + d_{\text{wind}}(h) \quad \text{with} \quad \begin{cases} d_{\text{bottom}}(h, \mathbf{u}) = -c_{\text{bottom}} \dfrac{|\mathbf{u}|\mathbf{u}}{\rho h} \\[2mm] d_{\text{wind}}(h) = \dfrac{\tau_{\text{wind}}}{\rho h} \end{cases}$$



with $f_{\text{bottom}}(h, \mathbf{u})$ denoting a forcing term due to the bottom drag and $f_{\text{wind}}(h)$ denoting a forcing term due to wind drag. Here, $c_{\text{bottom}}$ denotes the bottom drag coefficient chosen to be $10^{-3}$, $\rho$ denotes the density, and $\tau_{\text{wind}}$ denotes the surface wind stress. See, e.g., Wolfram et. al. (2015) for a discussion of this choice for $G(h, \mathbf{u})$.

For constructing the MC and MFMC estimators, we use three SCVT grids of the SOMA domain $\Gamma$ given by

| grid resolution | | number of cells | number of edges | number of vertices |
|---|---|---|---|---|
| 8km | $\Rightarrow$ | 120,953 | 364,124 | 243,172 |
| 16km | $\Rightarrow$ | 30,217 | 91,285 | 61,069 |
| 32km | $\Rightarrow$ | 8,521 | 25,898 | 17,378 |

A representative SCVT meshing of $\Gamma$ is illustrated in Figure 1. Note that for the SCVT grids used by the TRiSK scheme, the cells are almost all hexagonal, with a few pentagons and heptagons thrown in.

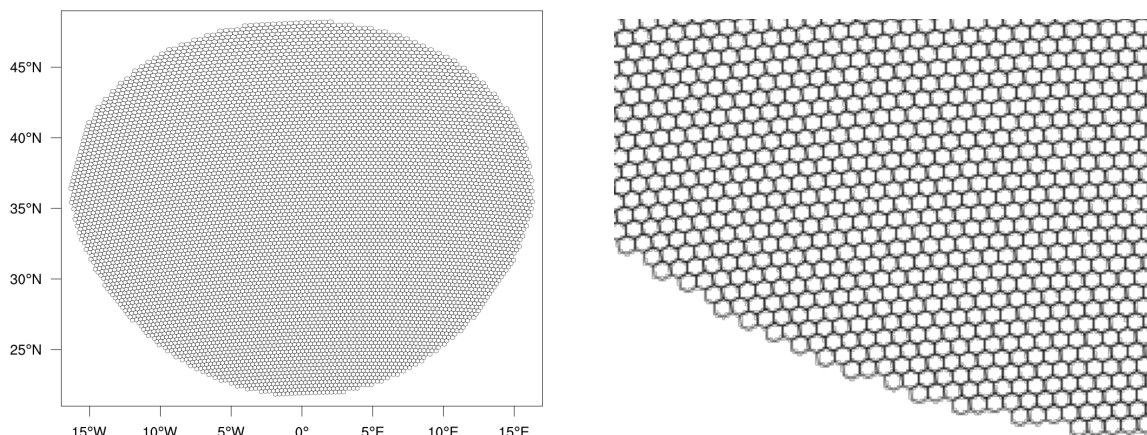

**Figure 1.** The 32km grid used for the SOMA test case and a zoom-in on a portion of that mesh.

The *output of interest* we consider is the maximum sea-surface height which is a common benchmark in oceanic RSWE simulations and which is relevant to, e.g., the detection of phenomena such as flooding. In particular, we simulate the RSWE until the final time of $T = 3$ days using the finest grid (8km resolution) of $120,953$ cells and then choose the OoI $F^1_{soma}$ given

by

$$F^1_{soma} = \max_{\ell=1,\dots,120,953} h_\ell, \tag{13}$$

where $h_\ell$ denotes the sea-surface height (also called fluid thickness) at the center of the cell $P_\ell$ at the final time $T = 3$.

The *quantity of interest* we choose considers the effect that perturbations of the initial velocity $\mathbf{u}_0 = \mathbf{u}(0)$ have on $F^1_{soma}$ and how that effect can be quantified using MC and MFMC estimation. To this end, consider random dilations $(1 + z)\mathbf{u}_0$ of the

initial velocity depending on the i.i.d. random variable $z$ that is uniformly distributed over the interval $[-0.5, 0.5]$. Then, the





OoI defined in (13) depends on the choice of $z$ in that interval, i.e., we have that $F_{soma}^1 = F_{soma}^1(z)$. In particular, we choose the QoI to be the expectation $Q_{1,soma} = \mathbb{E}\left[F_{soma}^1\right]$ of the output of interest $F_{soma}^1(z)$ defined in (13).

Note that in practical RSWE simulations, as is the case for more sophisticated models such as the primitive equations, an approximation of the initial data $\mathbf{u}_0$ is often obtained from a pre-processing procedure in which the RSWE system is *spun-up*

from rest, i.e., from a zero initial condition for $\mathbf{u}$, up to some specified time; see Anderson et. al. (1975); Bleck and Boudra (1986). This procedure is invoked so as to eliminate transient artifacts which are not present in the current ocean or atmosphere and produces an initial configuration which is closer to observed oceanic and atmospheric data. The outcome of the spin-up calculation over the spin-up time frame of 15 days is illustrated in Figure 2. The initial conditions $\mathbf{u}_0$ and $h_0$ that supplement (12) are then simply set to the outcome of this pre-processing step, i.e., to the sea-surface height and velocity obtained at the end of the spin-up calculation.

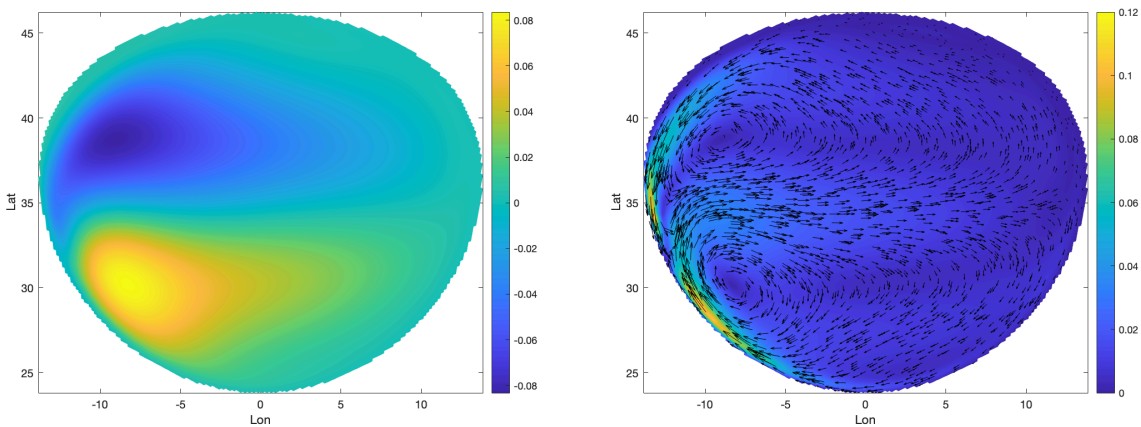

**Figure 2.** For the setting of Section 3.1, the RSWE truth model solution (8km resolution) with thickness $h$ (left) and velocity field $\mathbf{u}$ (right) after integration of the system from rest for $T = 15$ days.


### 3.1.1 MC and MFMC estimators

The MC estimator $Q_{1,soma}^{MC}$ of $Q_{1,soma} = \mathbb{E}\left[F_{soma}^1\right]$ is given by (2). Unfortunately, obtaining acceptable accuracy using a MC estimator suffers from a double shortcoming. First, for any given $z$, obtaining $F_{soma}^1$ is a costly endeavor because it requires the solution of the discretized RSWE system to obtain the necessary 120,953 values of $h_\ell(z)$. Second, to obtain an MC estimator

that is acceptable accuracy requires obtaining $F_{soma}^1$ for many sample values of $z$.

Naturally, to mitigate this double shortcoming, we turn to the MFMC estimator described in box **B** of Section 2. To do so, we define three surrogate OoIs $F_{soma}^2(z)$, $F_{soma}^3(z)$, and $F_{soma}^4(z)$, all three of which are less costly to obtain compared to that of $F_{soma}^1(z)$. Here, $F_{soma}^2(z)$ and $F_{soma}^3(z)$ are simply based on solving the discretized RSWE system using the coarser 16km and 32km grids, respectively. The third surrogate $F_{soma}^4(z)$ is the piecewise-linear interpolant based on the values of

$F_{soma}^1(z)$ which is exact at the three points $z = \{-0.5, 0, 0.5\}$. These four OoIs constitute a reasonable multifidelity ensemble





of models which are available during computational ocean studies, even more realistic ones such as primitive equation models. Moreover, all four OoIs can be leveraged for MFMC estimation whereas more traditional MC and multilevel MC estimators only make use of the first or the first three OoIs, respectively. Approximations of the costs and correlations for the four OoIs $\{F_{soma}^k(z)\}_{k=1}^4$ are obtained by considering 100 uniform i.i.d. samples of $z \in [-0.5, 0.5]$. These are computed to be

$$\mathbf{C} = \begin{pmatrix} C_1 \\ C_2 \\ C_3 \\ C_4 \end{pmatrix} = \begin{pmatrix} 101.1 \\ 12.83 \\ 1.714 \\ 0.05 \end{pmatrix} \quad \text{and} \quad \boldsymbol{\zeta}_1 = \begin{pmatrix} \zeta_{1,1} \\ \zeta_{1,2} \\ \zeta_{1,3} \\ \zeta_{1,4} \end{pmatrix} = \begin{pmatrix} 1.00000000 \\ 0.99975045 \\ 0.99975920 \\ 0.99974835 \end{pmatrix}, \tag{14}$$

where $C_k$ for $k = 1, 2, 3$ denote the average computation time (in wall-clock seconds) necessary to advance the relevant discretized RSWE system one time step, computed using 500 time steps for the simulation parameter $z = 0.001$. Note that the cost $C_4$ is assigned arbitrarily because the cost of evaluating the interpolant is negligible. These cost-correlation pairs are ideal for MFMC estimation, i.e., the surrogates are very well correlated with $F_{soma}^1(z)$ and are much less costly to obtain compared
to $F_{soma}^1(z)$. Note that the 100 samples of $z$ used to determine the data in (14) can be reused when determining the MC and MFMC estimators.

From (14), we observe that the more expensive surrogate $F_{soma}^2(z)$ is slightly less correlated to $F_{soma}^1(z)$ than is the cheaper surrogate $F_{soma}^3(z)$. So, to satisfy the first criteria in (9), the correlations should be ordered in the decreasing order $\zeta_{1,1}$, $\zeta_{1,3}$, $\zeta_{1,2}$, and $\zeta_{1,4}$. However, it turns out that then the second criteria in (9) for $k = 2$ is not satisfied so that the $F_{soma}^2(z)$ is removed
from the list of surrogates when computing the MFMC estimator. That estimator is given by (4) with $K = 4$ and where $k = 2$ is excluded from the sum. This is an example which illustrates, as discussed in Section 2, that not all surrogates one chooses contribute to the efficiency of MFMC estimation and, on the other hand, their omission does no harm to the accuracy of that estimator.

As already alluded to, the goal is to compare, for the same budget, the MSEs of the approximations $Q_{soma}^{MC}$ and $Q_{soma}^{MFMC}$ to
the exact QoI $Q_{1,soma} = \mathbb{E}[F^1]$ produced by the MC and MFMC estimators. However, there is still an additional challenge to be dealt with, namely that it is unclear how to choose a reference QoI that one can use to determine these MSEs, as only limited data (200 high-fidelity samples) are available and an MC approximation $Q_{soma}^{MC}$ with 200 samples is not sufficient for an accurate estimate of the expectation $\mathbb{E}[F^1]$. Therefore, we choose $Q_{soma}^{ref}$ to denote the average of the MFMC approximation $Q_{soma}^{MFMC}$ at the highest budget $B = 128C_1$ taken over 250 runs which use samples $z$ that are independently drawn from the
interval $[-0, 5, 0.5]$ *except for the fact* that the first $M_3$ (recall that $M_2$ is omitted) samples of each run are drawn from the pre-collected sampling set of size 200. The use of MFMC in defining $Q_{soma}^{ref}$ makes use of the fact that the MFMC method is at least as accurate as MC estimation, which can be verified through the inequality (7). With this choice, the MSEs in the MFMC and MC estimators can be measured with respect to the "exact quantity" $Q_{soma}^{ref}$.

The results of this procedure are given in Table 1 and Figure 3 from which it is evident that MFMC produces a much
more precise estimate compared to that of MC. In addition to MSE $e(Q_{soma}^{MFMC})$, we report the relative MSE defined as $e_{rel}(Q_{soma}^{MFMC}) = e(Q_{soma}^{MFMC})/(Q_{soma}^{ref})^2$. It is interesting that most of the computational budget is loaded onto the very crude piecewise-linear interpolant approximation $F_{soma}^4(z)$, allowing MFMC to achieve not only a lower MSE but also an estimate



with much smaller variance from run-to-run. Especially telling is the bottom plot in Figure 3 from which it is obvious that, for
the same budget, MFMC estimation results in greater accuracy of 2 or 3 orders of magnitude compared to that MC estimation,
or looking at it another way, for the same relative MSE, MFMC estimation requires a much smaller budget compared to MC
estimation.

**Table 1.** Results of the SOMA RSWE test with perturbed initial velocities for budgets $B = 2^k C_1$ equivalent to $2^k$ highest-fidelity runs.

| MC | | |
|---|---|---|
| $k$ | $e(Q_{soma}^{MC}) \times 10^{-5}$ | $e_{rel}(Q_{soma}^{MC}) \times 10^{-3}$ |
| 2 | 22.69 | 25.89 |
| 4 | 11.08 | 12.65 |
| 8 | 6.086 | 6.946 |
| 16 | 2.539 | 2.898 |
| 32 | 1.167 | 1.331 |
| 64 | 0.4156 | 0.4743 |
| 128 | 0.1306 | 0.1491 |

| Modified MFMC | | | | | |
|---|---|---|---|---|---|
| $k$ | number of samples taken of | | | $e(Q_{soma}^{MFMC})$ | $e_{rel}(Q_{soma}^{MFMC})$ |
| | $F_{soma}^1(z)$ | $F_{soma}^3(z)$ | $F_{soma}^4(z)$ | $\times 10^{-8}$ | $\times 10^{-6}$ |
| 2 | 1 | 1 | 1968 | 43.09 | 49.18 |
| 4 | 1 | 3 | 4016 | 32.88 | 37.52 |
| 8 | 3 | 6 | 8032 | 10.50 | 11.98 |
| 16 | 7 | 12 | 16064 | 5.949 | 6.790 |
| 32 | 15 | 25 | 32128 | 2.483 | 2.833 |
| 64 | 31 | 51 | 64256 | 1.107 | 1.263 |
| 128 | 62 | 102 | 128512 | 0.5733 | 0.6542 |

## 3.2  Test Case 5 for the RSWE system

We again consider the RSWE system (12) but now $\Omega$ denotes the whole surface of the sphere. Specifically, we consider the
configuration of Test Case 5 defined in Williamson et. al. (1992), which is widely used as a benchmark and employed as
a stepping stone towards more realistic atmospheric models; see, e.g., Leng et. al. (2019); Meng et. al. (2020); Ringler et.
al. (2010); Williamson et. al. (1992). Note that for this example there are no additional forces acting on the system, so that
$G(h, \mathbf{u}) = 0$ in (12).





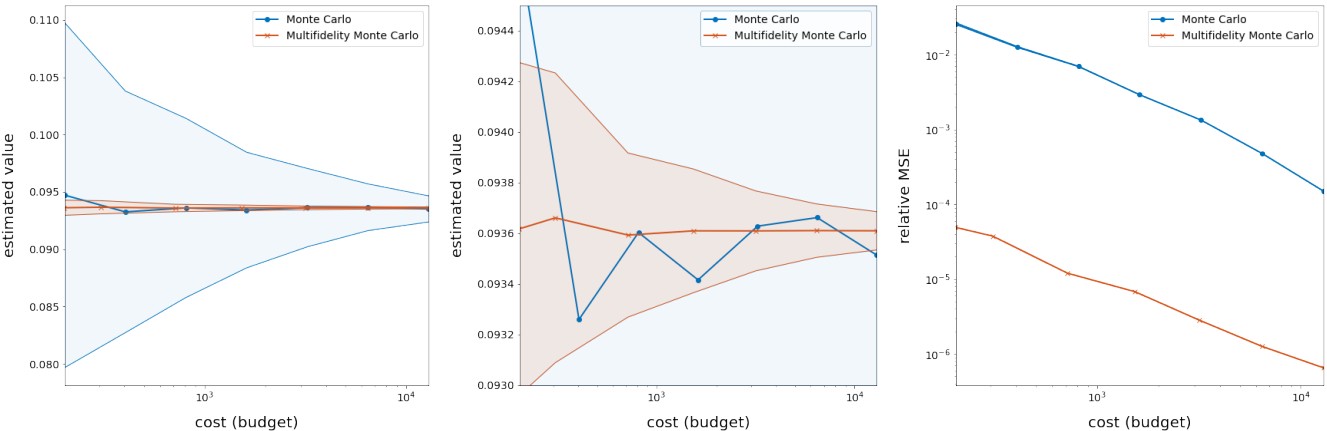

**Figure 3.** Results for the SOMA RSWE test with output of interest (13) averaged over 250 applications of MC and MFMC estimation. Top: the quantity of interest $Q$ as a function of the budget for the MC and MFMC estimators. Middle: a zoom-in of the top figure. Bottom: the relative MSE of the MC and MFMC estimators as a function of the budget; shadings represent the standard deviations of the MC and MFMC predictions when compared to their averages over the 250 runs.

Test Case 5 considers the flow over an isolated mountain centered at longitude $\lambda_c = \frac{3\pi}{2}$ and latitude $\theta_c = \frac{\pi}{6}$ with height

$$h_b(z_1) = z_1 \left( 1 - \frac{r}{a} \right), \tag{15}$$

where $a = \frac{\pi}{9}$, $r^2 = \min\{a^2, (\lambda - \lambda_c)^2 + (\theta - \theta_c)^2\}$, and $\lambda, \theta$ denote longitude and latitude, respectively. In (15), $z_1$ denotes a random variable that is uniformly distributed over the interval [1km, 3km].

The initial tangential (to the sphere) velocity in the longitudinal and latitudinal directions is chosen to be $\mathbf{u}_0(z_2) = (z_2 \cos\theta, 0)$, where $z_2$ denotes a random variable that is uniformly distributed over the interval [15m/s, 25m/s]. The initial sea-surface height $h$ is chosen as

$$h_0(z_2) = \widehat{h} - \frac{1}{g} \left( R\omega z_2 + \frac{z_2^2}{2} \right) \sin^2\theta,$$

where $\widehat{h} = 5.96$km, $R = 6371.22$km, and $\omega = 7.292 \times 10^{-5} s^{-1}$. With this, the solution $\mathbf{u}(\mathbf{z})$ and $h(\mathbf{z})$ of the RSWE system (12) depends on the random vector $\mathbf{z} = (z_1, z_2) \in [1\text{km}, 3\text{km}] \times [15\text{m/s}, 25\text{m/s}]$. In Figure 4 we provide an example of the initial thickness $h_0(z_2)$ and the thickness $h(z_1, z_2)$ after 10 days for specific values of $z_1$ and $z_2$.

For the simulation results given in Figure 4 and for other results in this subsection, we use the TRiSK scheme for spatial

discretization and a 4th-order explicit Runge-Kutta method for temporal discretization. SCVT uniform gridding is employed as in Section 3.1, although now the grid covers the whole surface of the sphere. A comparison of two such SVCT grids with 480 km and 240 km resolutions is provided in Figure 5. In constructing the MC and MFMC estimators, we use the three globally refined SCVT meshes of the whole sphere given by




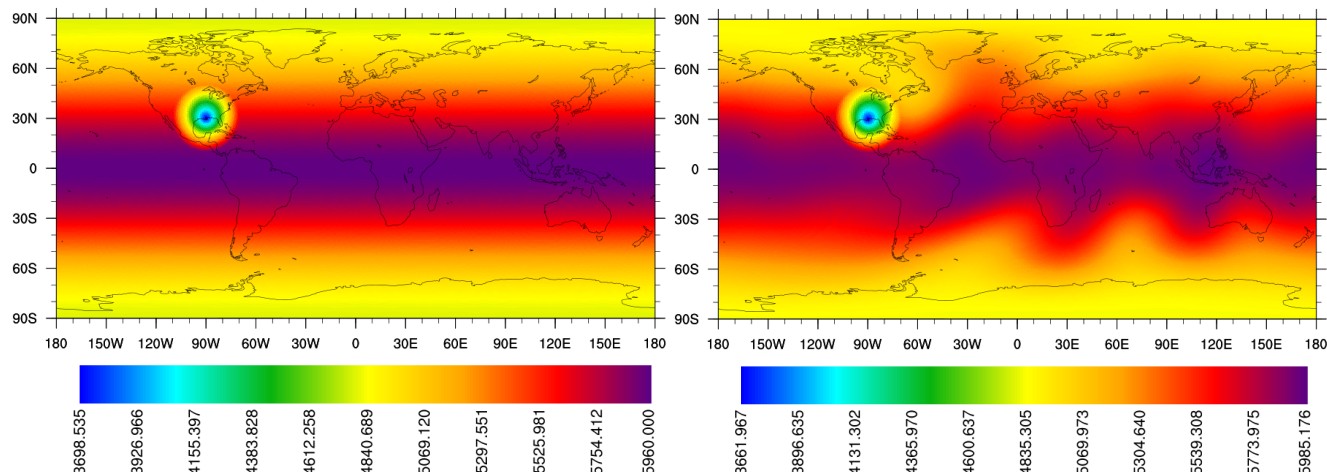

**Figure 4.** The initial thickness $h_0(z_2)$ with $z_2 = 23.82865$m (top) and the thickness $h$ after 10 days (bottom) with the mountain height $h_b(z_1)$ with $z_1 = 2023.78$m.

| grid resolution | | number of cells | number of edges | number of vertices |
|---|---|---|---|---|
| 120km | $\Rightarrow$ | 40,962 | 122,880 | 81,920 |
| 240km | $\Rightarrow$ | 10,242 | 30,720 | 20,480 |
| 480km | $\Rightarrow$ | 2,562 | 7,680 | 5,120 |

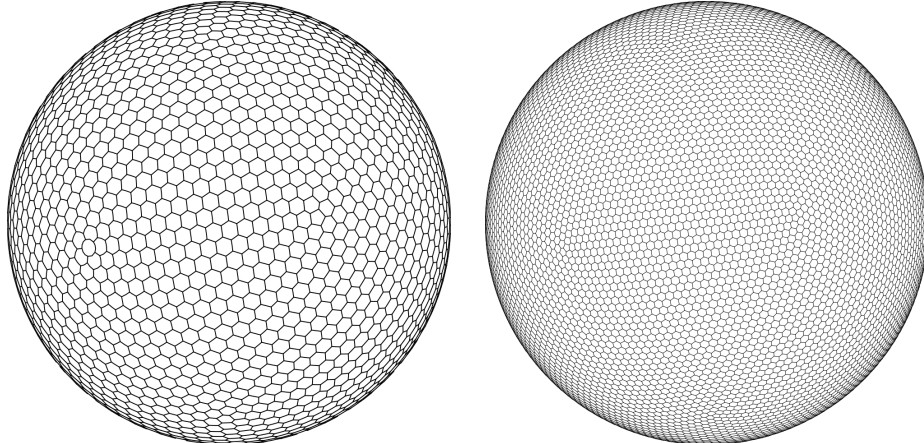

**Figure 5.** Two global SCVT meshes of the sphere surface with different grid resolutions: 480 km (left) and 240 km (right).





For the *output of interest*, we choose

$$F_{test5}^1(\mathbf{z}) = \frac{1}{N_\ell^1} \max_{\ell=1,\dots,N_\ell^1} |u_{e,\ell}(\mathbf{z})|,$$

where $N_\ell^1$ denotes the number of cell edges for the finest resolution case of 120km and $u_{e,\ell}(\mathbf{z})$ denotes the value of the normal component of velocity $u_e$ at the $\ell$-th cell edge for any choice of the random vector $\mathbf{z} = (z_1, z_2) \in [1\text{km}, 3\text{km}] \times [15\text{m/s}, 25\text{m/s}]$. We then have that the *quantity of interest* is given by

$$Q_{1,test5} = \mathbb{E}\left[F_{test5}^1(\mathbf{z})\right]. \tag{16}$$

### 3.2.1   MC and MFMC estimators

Similar to before, the goal is now to construct and compare, for the same computational budget, MC and MFMC estimators of the QoI defined in (16). The MC estimator $Q_{1,test5}^{MC}$ of the QoI $Q_{1,test5} = \mathbb{E}\left[F_{test5}^1\right]$ is given by (2). The MFMC estimator $Q_{1,test5}^{MFMC}$ of $Q_{1,test5} = \mathbb{E}\left[F_{test5}^1\right]$ makes use not only of the MC estimator $Q_{1,test5}^{MC}$ for the finest 120km grid, but also of the MC
estimators $Q_{2,test5}^{MC} \approx \mathbb{E}\left[F_{test5}^1\right]$ and $Q_{3,test5}^{MC} \approx \mathbb{E}\left[F_{test5}^1\right]$ corresponding the coarser 240km and 480km grids, respectively.

In this case, 4500 uniform i.i.d. realizations of $\mathbf{z}$ are drawn for pre-computation, of which all models share 1200 and models the two lower-fidelity surrogates share the entire 4500. Using 100 samples of the random variable $\mathbf{z}$, the costs and correlation coefficients of these models are respectively approximated by

$$\mathbf{C} = \begin{pmatrix} C_1 \\ C_2 \\ C_3 \end{pmatrix} = \begin{pmatrix} 434.8 \\ 126.9 \\ 58.04 \end{pmatrix} \quad \text{and} \quad \boldsymbol{\zeta}_1 = \begin{pmatrix} \zeta_{1,1} \\ \zeta_{1,2} \\ \zeta_{1,3} \end{pmatrix} = \begin{pmatrix} 1.00000000 \\ 0.99986604 \\ 0.99925882 \end{pmatrix},$$

where the costs $C_1$, $C_2$, and $C_3$ denote the computation time (in wall-clock seconds) necessary to advance the relevant discretized RSWE system one time step. As was the case for the SOMA experiment, these cost-correlation pairs are ideal for MFMC estimation, i.e., the surrogates OoIs $F_{test5}^2(\mathbf{z})$ and $F_{test5}^3(\mathbf{z})$ are very well correlated with $F_{test5}^1(\mathbf{z})$ and are much less costly to obtain compared to $F_{test5}^1(\mathbf{z})$. Note the 100 samples of $\mathbf{z}$ used to determine the data in (14) can be reused when determining the MC and MFMC estimators. From this point on, all other details about the construction and use of the MC and
MFMC estimators are the same as for the SOMA test case of Section 3.1.

The results provided in Table 2 and Figure 6 show that MFMC estimation produces a much more precise estimate compared to MC estimation. As was true for the SOMA test case, especially telling is the bottom plot in Figure 6 from which it is obvious that, for the same budget, MFMC estimation results in an order of magnitude greater accuracy compared to that MC estimation, or looking at it another way, for the same relative MSE, MFMC estimation requires a much smaller budget compared to MC
estimation.

## 4   First-order ice sheet model

The next experiment we consider illustrates the effectiveness of MFMC estimation on a QoI important for the realistic modeling of ice sheets such as those found near, e.g., Greenland, Antarctica, and various glaciers.





**Table 2.** Results of Test Case 5 with perturbed initial velocities for budgets $B = 2^k C_1$ equivalent to $2^k$ high-fidelity runs.

| | MC | |
|---|---|---|
| $k$ | $e(Q_{test5}^{MC})$ | $e_{rel}(Q_{test5}^{MC}) \times 10^{-3}$ |
| 2 | 26.95 | 18.51 |
| 4 | 14.54 | 9.987 |
| 8 | 6.884 | 4.729 |
| 16 | 3.689 | 2.534 |
| 32 | 1.609 | 1.105 |
| 64 | 0.7501 | 0.5153 |
| 128 | 0.2846 | 0.1955 |
| 256 | 0.1048 | 0.07202 |

| | Modified MFMC | | | | |
|---|---|---|---|---|---|
| $k$ | number of samples taken of | | | $e(Q_{test5}^{MFMC})$ | $e_{rel}(Q_{test5}^{MFMC})$ |
| | $F_{test5}^1(z)$ | $F_{test}^2(z)$ | $F_{test}^3(z)$ | $\times 10^0$ | $\times 10^{-3}$ |
| 2 | 1 | 1 | 5 | 11.49 | 7.891 |
| 4 | 1 | 1 | 20 | 2.838 | 1.950 |
| 8 | 1 | 1 | 49 | 1.147 | 0.7879 |
| 16 | 1 | 2 | 106 | 0.6137 | 0.4216 |
| 32 | 1 | 5 | 218 | 0.2927 | 0.2011 |
| 64 | 2 | 10 | 437 | 0.09758 | 0.06704 |
| 128 | 5 | 20 | 874 | 0.03788 | 0.02603 |
| 256 | 10 | 41 | 1748 | 0.01169 | 0.008034 |

## 4.1 The first-order model for ice sheets

The dynamical behavior of ice sheets is commonly modeled by what is referred to as the *first-order model* or the *Blatter–Pattyn model*. Here, we provide a short review of that model; detailed descriptions are given in, e.g., Blatter (1995); Pattyn (2003) and also in Perego et. al. (2012); Tezaur et. al. (2015).

Let $\Omega$ denote the three-dimensional domain occupied by the ice sheet having boundary $\Gamma = \Gamma_s \cup \Gamma_b \cup \Gamma_\ell$ given by

$$\begin{aligned}
\Gamma_s &\Leftarrow x_3 = s(x_1, x_2) &\Rightarrow \text{top surface of the ice sheet} \\
\Gamma_b &\Leftarrow x_3 = b(x_1, x_2) &\Rightarrow \text{bottom (or basal) surface of the ice sheet} \\
\Gamma_\ell &\Leftarrow \ell(x_1, x_2) = 0 &\Rightarrow \text{lateral boundary.}
\end{aligned} \tag{17}$$

Figure 7 provides an illustration of the boundary segments defined in (17).





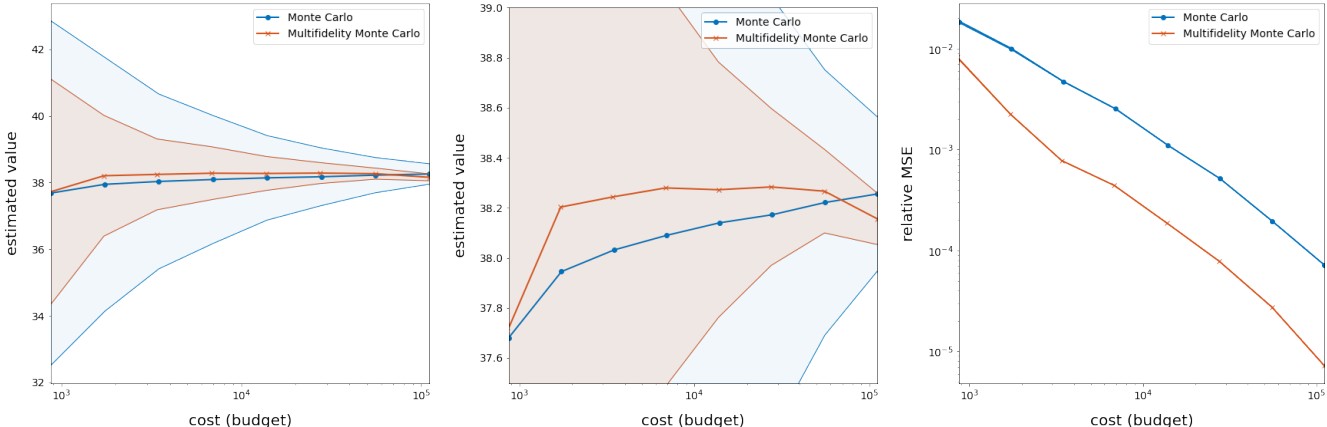

**Figure 6.** Results over 250 runs of the RSWE system for the Test Case 5 experiment with quantity of interest given by 16. Shading represents variance in the MC and MFMC predictions over the 250 runs, which are independent except for the fact that all random sampling employs the same pre-collected (random) set of parameters.

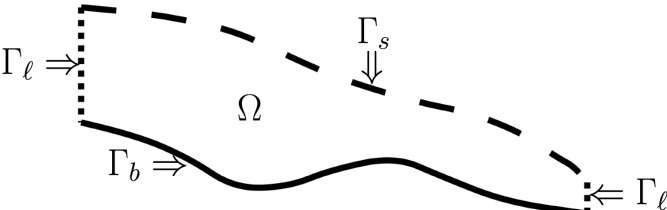

**Figure 7.** An $(x_1, x_3)$ cross section of the three-dimensional domain $\Omega$ occupied by an ice sheet and the boundary segments defined in (17).

Also, let $u_1$ and $u_2$ denote the $x_1$ and $x_2$ components of the velocity vector $\mathbf{u} = (u_1 \ u_2 \ u_3)^\mathsf{T}$. Then, the first-order model equations for ice sheets is given by the partial differential equations

$$
\begin{cases}
-\nabla \cdot (2\mu\boldsymbol{\epsilon}_1) + \rho g \dfrac{\partial s}{\partial x_1} = 0 \\[2mm]
-\nabla \cdot (2\mu\boldsymbol{\epsilon}_2) + \rho g \dfrac{\partial s}{\partial x_2} = 0
\end{cases}
\qquad \text{for } \mathbf{x} = (x_1 \ x_2 \ x_3)^\mathsf{T} \in \Omega, \tag{18}
$$

where $\mu$ denotes the viscosity coefficient, $g$ denotes the gravitational acceleration, and $\rho$ denotes the density. In (18), the strain-rate tensor $(\boldsymbol{\epsilon}_1, \boldsymbol{\epsilon}_2)$ is given by

$$
\boldsymbol{\epsilon}_1 = \begin{pmatrix} 2\dfrac{\partial u_1}{\partial x_1} + \dfrac{\partial u_2}{\partial x_2} \\[3mm] \dfrac{1}{2}\Big(\dfrac{\partial u_1}{\partial x_2} + \dfrac{\partial u_2}{\partial x_1}\Big) \\[3mm] \dfrac{1}{2}\dfrac{\partial u_1}{\partial x_3} \end{pmatrix}
\qquad \text{and} \qquad
\boldsymbol{\epsilon}_2 = \begin{pmatrix} \dfrac{1}{2}\Big(\dfrac{\partial u_1}{\partial x_2} + \dfrac{\partial u_2}{\partial x_1}\Big) \\[3mm] \dfrac{\partial u_1}{\partial x_1} + 2\dfrac{\partial u_2}{\partial x_2} \\[3mm] \dfrac{1}{2}\dfrac{\partial u_2}{\partial x_3} \end{pmatrix}
$$





and the nonlinear viscosity coefficient $\mu$ is given by the Glen flow law

$$\mu = \frac{1}{2}A^{-\frac{1}{n}}\boldsymbol{\epsilon}_e^{\frac{1}{n}-1}$$

with $n = 3$ being the usual choice. The effective strain rate $\boldsymbol{\epsilon}_e$ is given by

$$\epsilon_e^2 = \left(\frac{\partial u_1}{\partial x_1}\right)^2 + \left(\frac{\partial u_2}{\partial x_2}\right)^2 + \frac{\partial u_1}{\partial x_1}\frac{\partial u_2}{\partial x_2} + \frac{1}{4}\left(\frac{\partial u_1}{\partial x_2} + \frac{\partial u_2}{\partial x_1}\right)^2 + \frac{1}{4}\left(\frac{\partial u_1}{\partial x_3}\right)^2 + \frac{1}{4}\left(\frac{\partial u_2}{\partial x_3}\right)^2,$$

and $A$ is often chosen to obey the Arrhenius relation

$$A = A(T) = a\exp(-Q/RT),$$

where $T$ denotes the absolute temperature measured in degrees Kelvin, $R$ denotes the universal gas constant, $Q$ denotes the activation energy for creep, and $a$ is an empirical flow constant often used as a tuning parameter.

The system (18) is supplemented by the boundary conditions

$$
\begin{array}{llll}
\boldsymbol{\epsilon}_1 \cdot \mathbf{n} = 0 & \text{and} & \boldsymbol{\epsilon}_2 \cdot \mathbf{n} = 0 & \text{on } \Gamma_s \\
2\mu\boldsymbol{\epsilon}_1 \cdot \mathbf{n} + \beta u = 0 & \text{and} & 2\mu\boldsymbol{\epsilon}_2 \cdot \mathbf{n} + \beta v = 0 & \text{on } \Gamma_b \\
u = 0 & \text{and} & v = 0 & \text{on } \Gamma_\ell,
\end{array}
\tag{19}
$$

where $\beta$ denotes a basal friction parameter.

Once the horizontal components $u_1$ and $u_2$ of the velocity are determined, the vertical velocity component $w$ is determined by enforcing incompressibility, i.e., we have that

$$\frac{\partial w}{\partial x_3} = -\frac{\partial u_1}{\partial x_1} - \frac{\partial u_2}{\partial x_2} \qquad \text{for } \mathbf{x} \in \Omega. \tag{20}$$

Because the right-hand side is known, this is an ordinary differential equation for $w$.

Of paramount interest in the modeling of ice sheets is the monitoring of the *temporal evolution* of the ice sheet domain $\Omega$. However, one notices that there are no time derivatives of $u_1$ and $u_2$ appearing in (18), i.e., that system is a static one. The reasoning behind this curiosity is twofold. Firstly, we have that the system (18) is coupled to an equation for the temporal

evolution of the temperature $T$ within the ice sheet, and also coupled to an equation for the temporal evolution of the top surface of the ice-sheet which engenders changes in the ice-sheet domain $\Omega$. Secondly, the time scale of changes in the temperature is much shorter (e.g., hourly) when compared to the time scale (e.g., at least many days) of changes in the ice sheet domain $\Omega$.

Thus, in a computational model of ice-sheet dynamics, determination of the domain $\Omega$ is coupled to that of the velocity $\mathbf{u}$. Precisely, the temperature and top surface are first advanced from a given domain/velocity pair $(\Omega, \mathbf{u})$ over several time

steps, producing an evolution that prescribes a new domain/temperature pair $(\Omega, T)$. Following this, the pair $(\Omega, T)$ are used to generate a new velocity $\mathbf{u}$ by solving (18) and (20). While both stages of this process are important, the present experiment focuses on the computation of $\mathbf{u}$ from $\Omega$ and $T$.



## 4.2 MC and MFMC estimation

The specific application setting we consider is based on Experiment C of the benchmark examples in Leng et. al. (2012); Pattyn
et. al. (2008). Here, the ice domain is a rectangular parallelepiped having a square base with side length $L$ and thickness $H$,
which lies on a slanted bed with slope parameter $\theta$. The upper surface boundary is given as

$$z_s(x_1, x_2) = -x_1 \tan\theta,$$

and the basal topography is given as

$$z_b(x_1, x_2) = z_s(x_1, x_2) - H.$$

For constructing the MC and MFMC estimators, we set the length $L = 80$km and height $H = 1$km and then use uniform
tetrahedral grids of the ice domain $\Omega_t$ with 120, 60, and 30 grid intervals in each horizontal direction and, correspondingly, 20,
10, and 5 intervals in the vertical direction. As a result, we have that number of vertices, horizontal triangles, and tetrahedra
are determined to be

| grid resolution | | number of vertices | number of horizontal triangles | number of tetrahedra |
|---|---|---|---|---|
| $120 \times 120 \times 20$ | $\Rightarrow$ | 307,461 | 28,800 | 1,728,000 |
| $60 \times 60 \times 10$ | $\Rightarrow$ | 40,931 | 7,200 | 216,000 |
| $30 \times 30 \times 5$ | $\Rightarrow$ | 5,766 | 1,800 | 27,000 |

The first-order ice-sheet model is discretized using the stabilized P1-P1 finite elements given in Zhang et. al. (2011). To
solve the resulting nonlinear system of discrete equations, 15 Picard iteration steps are carried out after which a switch is made
to a Newton iteration, up to a maximum of a total of 40 iterations. However, if the residual error decrease is less than 75%
relative to the residual error of the previous step, the nonlinear iteration is switched back to a Picard iteration. An example
illustration of the discrete solution of this ice-sheet model for specific values of the parameters $\theta$ and $\beta$ is provided in Figure 8
using the highest resolution grid.

Because the cracking and melting of ice sheets is an important indicator of climate change (Clark et. al. (1999); Hanna et.
al. (2013)), we consider the *the output of interest*

$$F_{ice}^1(\mathbf{z}) = \frac{1}{2N_1} \sum_{\ell=1}^{N_1} |\mathbf{u}_n(\mathbf{z})|^2,$$

where $N_1 = 307,461$ is the number of vertices of the finest $120 \times 120 \times 20$ grid, $\mathbf{u}_n(\mathbf{z})$ denotes the value of the discrete velocity
$\mathbf{u}$ at the $n$th vertex for any choice of the i.i.d. random vector $\mathbf{z} = (\theta, \beta) \in [0.2, 0.8] \times [800\text{m}, 1200\text{m}]$. This OoI provides a
measurement of how energetic the ice sheet is depending on its slope and friction coefficient, and can be loosely related to how
vigorously the ice will deform given a configuration specified by $\mathbf{z}$. We then choose the *quantity of interest* given by

$$Q_{1,ice} = \mathbb{E}\left[F_{ice}^1(\mathbf{z})\right]. \tag{21}$$





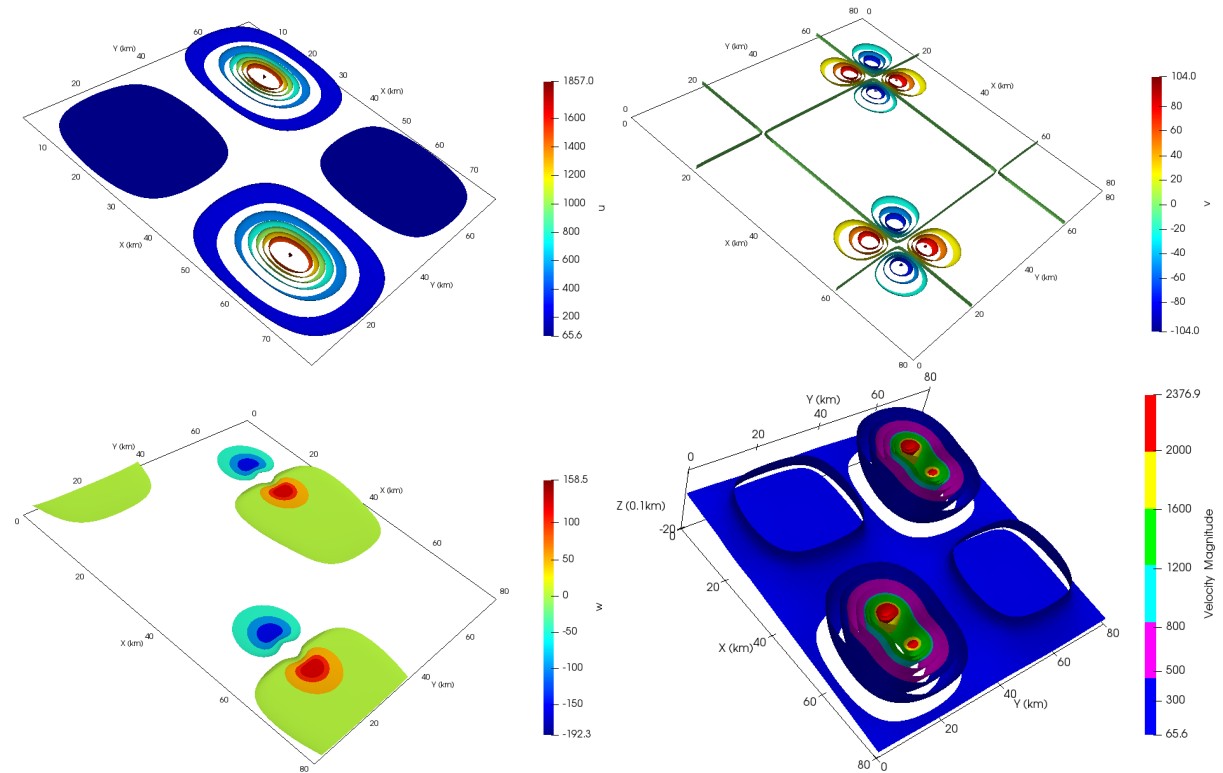

**Figure 8.** A simulation of the ice-sheet model with $\theta = 0.7387$ and $\beta = 901.5232$, discretized at the highest grid resolution. The top two plots display contours of the velocity components $u_1$ and $u_2$, respectively, on the top surface $\Gamma_s$. The third plot displays a contour plot of velocity component $u_3$. The fourth plot displays a cutaway plot of the velocity magnitude $|\mathbf{u}|$ inside the sheet.

Two surrogates for the ice-sheet model

$$
F_{ice}^2(\mathbf{z}) = \frac{1}{2N_2} \sum_{n=1}^{N_2} |\mathbf{u}_n(\mathbf{z})|^2 \qquad \text{and} \qquad F_{ice}^3(\mathbf{z}) = \frac{1}{2N_3} \sum_{n=1}^{N_3} |\mathbf{u}_n(\mathbf{z})|^2
$$

are defined respectively using the coarser $60 \times 60 \times 10$ grid with $N_2 = 40,931$ and the even coarser $30 \times 30 \times 5$ grid with $N_3 = 5,766$. Then, from (3), we have the corresponding three Monte Carlo estimators $\{Q_{1,ice}^{MC}, Q_{2,ice}^{MC}, Q_{3,ice}^{MC}\}$ and, from (4), we have the MFMC estimator $Q_{ice}^{MFMC}$ which makes use of these three MC estimators.

At this point, we proceed as was done in Section 3.2. For example, we now have that the approximate costs (measured in wall-clock seconds and averaged over 100 random samples) for computing $F_{ice}^1(\mathbf{z})$, $F_{ice}^2(\mathbf{z})$, and $F_{ice}^3(\mathbf{z})$, as well as the approximate correlations for the three OoIs are given by

$$
\mathbf{C} = \begin{pmatrix} C_1 \\ C_2 \\ C_3 \end{pmatrix} = \begin{pmatrix} 285.5 \\ 23.05 \\ 2.690 \end{pmatrix} \qquad \text{and} \qquad \boldsymbol{\zeta}_1 = \begin{pmatrix} \zeta_{1,1} \\ \zeta_{1,2} \\ \zeta_{1,3} \end{pmatrix} = \begin{pmatrix} 1.00000000 \\ 0.99999796 \\ 0.99996691 \end{pmatrix}.
$$





The remaining experimental details are nearly identical to those of Section 3.2. As was the case for the oceanic simulations, the computational expense of these models precludes the collection of unlimited data, so a total of 8000 uniform i.i.d. realiza-

tions of $\mathbf{z}$ are drawn of which all models share 200 and the lower-fidelity surrogates share 1000. The results of carrying out MFMC and MC estimation over 250 "independent" runs using this common data set are provided in Table 3 and Figure 9. Here it is especially apparent that the budget-preserving modifications to MFMC that led to the algorithm in Box **B** were necessary, as $B \ll \mathbf{C} \cdot \mathbf{R}$ and only one high-fidelity run is ever selected by the algorithm.

Despite this, it is clear from Table 3 and the plots in Figure 9 that, at these small budgets, MFMC estimation produces an

estimator which is much more accurate and precise than does MC estimation. This is not surprising because MFMC estimation has the flexibility to load most of its budget onto the cheaper lower-fidelity surrogate models, evaluating them many times in order to bring down the overall variance of their estimates. Again, this provides empirical validation for the use of MFMC estimation over MC estimation when estimating model statistics even in practical cases where data availability is low.

**Table 3.** Results of the ice-sheet experiment for budgets equivalent to $2^k$ high-fidelity runs.

| MC | | |
|---|---|---|
| $k$ | $e(Q_{ice}^{MC}) \times 10^6$ | $e_{rel}(Q_{ice}^{MC}) \times 10^{-2}$ |
| 2 | 37.26 | 11.79 |
| 4 | 20.08 | 6.354 |
| 8 | 11.62 | 3.676 |
| 16 | 5.296 | 1.676 |
| 32 | 2.457 | 0.7774 |
| 64 | 1.018 | 0.3221 |

| Modified MFMC | | | | |
|---|---|---|---|---|
| $k$ | number of samples taken of | | $e(Q_{ice}^{MFMC})$ | $e_{rel}(Q_{ice}^{MFMC})$ |
| | $F_{ice}^1(z)$ | $F_{ice}^2(z)$ | $F_{ice}^3(z)$ | $\times 10^4$ | $\times 10^{-4}$ |
| 2 | 1 | 1 | 97 | 69.67 | 22.05 |
| 4 | 1 | 1 | 309 | 16.81 | 5.323 |
| 8 | 1 | 1 | 726 | 3.793 | 1.200 |
| 16 | 1 | 4 | 1555 | 1.370 | 0.4335 |
| 32 | 1 | 8 | 3215 | 1.123 | 0.3554 |
| 64 | 1 | 17 | 6506 | 0.3393 | 0.1074 |



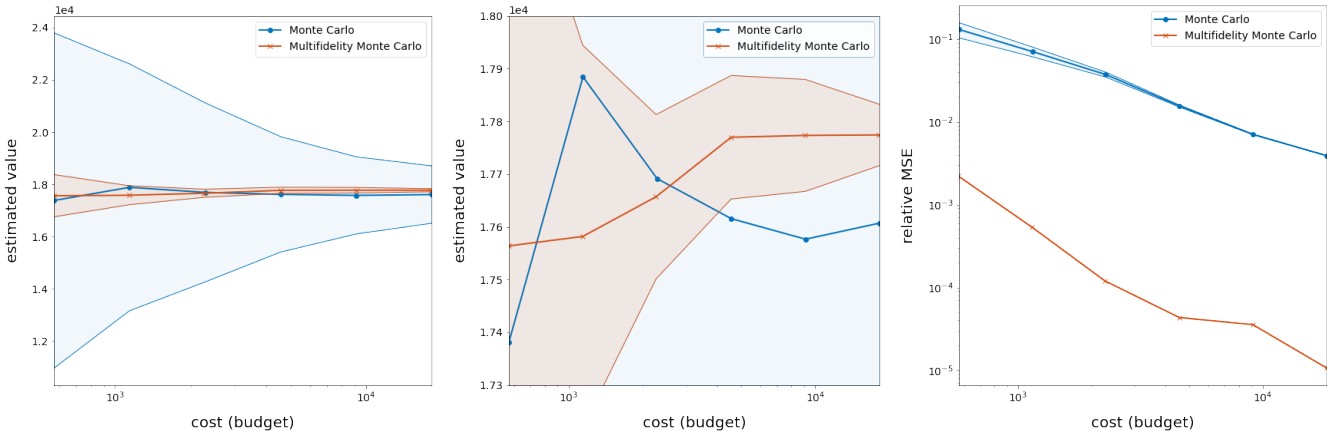

**Figure 9.** Results over 250 runs of the ice-sheet experiment with quantity of interest (21). Shading represents variance in the MC and MFMC predictions over the 250 runs.

## 5 Concluding Remarks

This paper serves to introduce multifidelity Monte Carlo estimation as an alternative to standard Monte Carlo estimation for quantifying uncertainties in the outputs of climate system models, albeit in very simplified settings. Specifically, we consider benchmark problems for the single-layer shallow water equations relevant to ocean and atmosphere dynamics and we also consider a benchmark problem for the first-order model of ice sheet dynamics. The computational results presented here are promising in that they amply demonstrate the superiority of MFMC estimation when compared to MC estimation on these

examples. Furthermore, the use of MFMC as an estimation method will surely be even more efficacious when quantifying uncertainties in more realistic climate modeling settings for which the simulation costs are prohibitively large, e.g., for long-time climate simulations. Thus, our next goal is to apply MFMC estimation to more useful models of climate dynamics (such as the primitive equations for ocean and atmosphere and the Stokes model for ice sheets) that are also coupled to the dynamics of other climate system components and also coupled to passive and active tracer equations.

*Code and data availability.* The climate simulation data used in this work along with Python code for reproducing the relevant experiments can be found at the Github repository Gruber et. al. (2022) with permanent identifier DOI:10.5281/zenodo.7071646.

*Author contributions.* The idea for this work was conceived by M. Gunzburger and the experiments presented were designed jointly by all authors. Software development, data collection, and figure generation were carried out by A. Gruber and R. Lan, while post-experiment analysis was conducted by all authors. The resulting manuscript was written by M. Gunzburger and A. Gruber with editorial contributions

from all authors.





*Acknowledgements.* This work is partially supported by U.S. Department of Energy Office of Science under grants DE-SC0020270, DE-SC0020418, and DE-SC0021077.





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
