# Peer review of "Multifidelity Monte Carlo Estimation for Efficient Uncertainty Quantification in Climate-Related Modeling"

_EGUsphere, 2022_

## Author Response (AR1)

Dear Prof. Lu,

Attached are our responses to the Anonymous Referees, along with a copy of the new manuscript with changes tracked.  We have listed the referee comments in blue for easier comparison.

Please let us know if we can do anything else to assist during the review process.

Thank you,
Anthony, Max, Rihui, Lili, and Zhu

**Response to Anonymous Referee #1:**

*In this paper, the authors extensively tested the multifidelity Monte Carlo (MFMC) method for accurately catching some quantities of interest in climate-related modeling. Compared to the widely used Monte Carlo (MC) method, they demonstrated that MFMC has superior properties over MC on this subject via three benchmark problems in the ocean, atmosphere, and ice sheet modeling. Since uncertainties exist almost everywhere in climate modeling and prediction, I think MFMC would greatly impact this area, especially in reducing the number of costly large-scale simulations in very fine resolutions. Hence, I would highly recommend the publication of this manuscript in GMD after the authors address the following minor issues.*

Thank you for your helpful comments and for your interest in our work.  Below are responses to your specific comments

1. *In line 67 on page 3, the statement is somehow confusing. The authors declare that they don't consider any possible alternate sampling schemes at the beginning. But their goal is to use a nontraditional MC sampling strategy. Doesn't the sampling strategy belong to the "alternate sampling schemes"?*

By "alternate sampling strategies" we mean that a user might be using (or may want to use) some other sampling scheme besides traditional MC for the truth model, e.g., Latin hypercube sampling, quasi-MC sampling, or sparse grid sampling, as doing so often results in lower costs for a given accuracy tolerance.  Therefore, the choice of sampling scheme is determined by how we want to sample the truth model and its surrogates.  While we use MC sampling in this work, other choices are possible and would produce analogues to the MFMC method, e.g., MFLH, MFQMC, or MFSG.

2. *In Algorithm A, it mentions that $F^k$ is removed from the list of surrogates if the second requirement is not satisfied. Will there be such a case all $F^k$ will be removed from the estimation? Secondly, will it obtain less accuracy compared to the MC method with all $F^k$ included?*

Note that all $F^k$ will never be removed according to our criterion; in the most extreme case, at least the high-fidelity model $F^1$ will remain.  On the other hand, if this occurs, we obtain the useful information that none of the surrogates $F^2, ..., F^k$ are effective at increasing estimation accuracy or reducing computational costs, i.e., we are stuck using MC on its own and should look for other, more informative, surrogates.  To address the other question, there should never be a loss of accuracy relative to MC regardless of the model set. The worst-case scenario is if all the surrogates fail the second test, in which case we are left with traditional MC using $F^1$.  Hence, there is no loss of accuracy

in MFMC; its worst-case accuracy is simply equal to that of MC using the truth model.   However, if at least one surrogate survives the second test, then the MFMC method guarantees that costs are reduced without compromising accuracy.

3.  For Figures 3 and 6, it is better to point out that the blue shading is the standard deviation of the MC prediction and the melon for MFMC. Moreover, being included in the blue shading indicates a smaller standard deviation.

We agree that it is a good idea to mention this explicitly.  Any future versions of the manuscript will include this change.

This change has been reflected in the new manuscript in the captions of Figures 3 and 6.

**Response to Anonymous Referee #2:**

*The authors describe a "multi-fidelity" Monte Carlo method for expected value estimation that is shown to improve convergence/efficiency compared to traditional Monte Carlo methods. This multi-fidelity approach is different to existing techniques in that expected values are computed using simulations run at differing resolutions and/or incorporating various levels of dynamical approximation. The test cases presented here employ hierarchies of progressively coarsened meshes to provide low-cost estimators, as well as an interpolation-based approach. The authors present results for two shallow-water benchmarks and one ice-sheet test-case, and demonstrate significant speed-ups compared to the traditional MC scheme for various maximum-value estimation problems. The speed-ups demonstrated for the MFMC method are very attractive, especially in the context of climate-model estimators which are often extremely expensive to obtain. There are a number of questions I have relating MFMC performance to problem nonlinearity though, in addition to various minor corrections.*

Thank you for your helpful comments and for your interest in our work.  Below are responses to your specific comments.

*- The three test-cases analysed appear to involve estimators evaluated after relatively short integration periods, and, rather than fully nonlinear flows/dynamics, the response of the system may be relatively linear over these short horizons as a result --- is this expected to influence the effectiveness of the MFMC results presented? For example, TC5 from Williamson does not become strongly nonlinear until approx. 20-days, with 50-days being the typical analysis window at which turbulence is fully developed. The SOMA case described in Wolfram (2015) is typically spun-up over several years, and then analysed over 30-day windows. In this work, it appears the TC5 case is analysed after 10 days, and the barotropic-gyre-version-of-SOMA after 3-days (restarted from a 15-day spin-up). Are the MFMC results robust when the duration/nonlinearity of the test-cases is increased?*

Based on our experience with prior work on MC-based estimation, we do not expect the linearity (or lack thereof) of the model solution to affect the performance of the MFMC estimation procedure.

*- Is it possible to estimate the relative "multi-fidelity" contributions to the accuracy of the MFMC estimator? For example, is the overall accuracy governed more by the small number of high resolution runs, the large number of low resolution runs, or something in-between? Considering the more linear (or at least non-turbulent) nature of solutions studied, how would a conventional MC estimator compare if run only using lower-resolution simulations? In other words, is the good performance of the MFMC method due to the solution being well-resolved even on the coarser meshes, or is the multi-resolution hierarchy effective in*

*estimating behaviour resolvable only at high-resolution? If it is the former, I wonder whether the problems studied are sufficiently nonlinear at the grid-scale. If it is the latter, this may be a nice result to highlight further.*

A goal of MFMC estimation is to achieve (using very few samples of the high-fidelity mode) the same accuracy as obtained by an MC estimator (that exclusively uses many samples of that model). Certainly, at least one high-fidelity model evaluation is necessary to eliminate bias in the MFMC estimator. Moreover, we find practically that the high-fidelity samples used to "steer" this estimator in an accurate direction, while the low-fidelity samples are used to shrink its variance around the true solution. Note that the dynamics of the example systems are not particularly well resolved by the low-fidelity models relative to the high-fidelity ones; particularly in the case of the barotropic gyre (SOMA) case, there is a noticeable visual difference between the 32km solution and the 8km solution.

*- I believe the gradient terms in the shallow-water system (12) should be grad(1/2*|u|^2) + g*grad(h + h_b) rather than the grad(rho) included currently. Here p = rho_0*g*h is used to simplify the linear 1/rho_0 * grad(p) shallow-water pressure gradient, consistent with e.g. Ringler et al (2010).*

This has now been corrected. Thank you for your attention.

This change is reflected in equation (12) in the new manuscript.

*- The SWE runtimes noted in 3.1.1 and 3.2.1 appear to be quite slow --- requiring 100's of seconds to advance a single time-step using relatively small O(<= 100,000) cell meshes? Are these runtimes for the full multi-day simulations instead, or for all ensemble members perhaps?*

These were the wall-clock times observed when the relevant system was implemented in MATLAB and run on a 2015 MacBook laptop. Therefore, neither the implementation nor the hardware was optimized for computational efficiency.

*- While the MFMC methods presented here are clearly different in that they leverage varying resolution simulations, is it fair to compare against only the "historical" MC method, which is known to be uncompetitive in terms of efficiency? Significant work on alternative MC methods has been conducted by various authors in which a variety of accelerated techniques have been proposed. Are the large gains reported for MFMC expected to be replicated compared to e.g. MCMC (Markov Chain Monte Carlo) approaches more frequently used in climate model estimation?*

A primary benefit of MFMC over other modern estimation methods such as MCMC is its ability to leverage low-fidelity information to effect cost-savings without sacrificing estimator accuracy.  In our experience with MFMC in other settings, this benefit translates to much larger cost savings for a given accuracy tolerance when compared to MCMC as well as other MC-related sampling schemes (e.g., variance reduction MC, importance sampling).

*Minor comments:*

*- The SOMA test case (Simulating Ocean Mesoscale Activity) typically refers to simulations using the multi-layer primitive equations, in which mesoscale eddies form due to 3d interactions between the momentum, density and forcing tendencies. In this shallow-water configuration with rho = const., it appears to be a wind-driven barotropic gyre that's studied instead, which is typically less turbulent, as per the smooth flow features in fig. 2. If so, it's suggested to label this test case as a wind-driven gyre.*

We agree with this reasoning and have changed the name globally throughout the manuscript.

This change is reflected throughout Section 3.1 in the new manuscript.

*- Wallis (2012) reference appears to be missing.*

This has been fixed, thank you for your attention.

This change is reflected in lines 185, 200-205, and 554 in the new manuscript.

*- ln 76: Is saying "no guesswork involved" too strong a statement? The systematic nature of the MFMC approach is attractive, but is it \*the\* provably optimal sampling strategy, or more of an effective heuristic?*

It can be shown that the MFMC method presented here is the provably optimal solution (up to rounding) to a particular constrained optimization problem (see Gruber et al 2022, "A multifidelity Monte Carlo method for realistic computational budgets", for a formal statement).  Therefore, we do not think it is a stretch to say there is "no guesswork involved" in this context.

*- ln 72: ...also uses cheaper to obtain...*

*- fig. 3 labelling: left-right vs top-bottom.*

These have been fixed, thank you for your attention.

These changes are reflected in (resp.) line 73 and the caption to Figure 3.

*- ln 308: Is this an expression for the free surface height or the layer thickness --- h appears to be thickness in the shallow-water system (12).*

This is an expression for the fluid thickness.  We have clarified this globally throughout the manuscript.

This change is reflected in Sections 3.1 and 3.2 of the new manuscript.

*- The Gruber (2022) paper referenced here appears to be an arXiv preprint, that in-turn references this GMD submission??*

This is true.  The referenced preprint (to appear in J. Sci. Comput.) establishes the particular MFMC algorithm which has been applied to climate-related examples in this paper.  Therefore, we refer interested readers to that manuscript for a more detailed description of the MFMC method.  Conversely, we write in that preprint that "Forthcoming work will investigate applications of the present MFMC method to complex systems governed by partial differential equations, particularly in the context of climate modeling.", and provide an empty citation with title and relevant authors.  The mentioned work has since become this GMD submission.

Since this has caused confusion, we intend to remove the offending citation from the J. Sci. Comput. paper during the proofing stage.

[revised manuscript text omitted]